

# Small-scale distribution of microbes and biogeochemistry in the Great Barrier Reef

Cátia Carreira[1], Júlia Porto Silva Carvalho[2], Samantha Talbot[3], Isabel Pereira[4] and Christian Lønborg[3,5]

[1] Department of Biology & CESAM –The Centre for Environmental and Marine Studies, University of Aveiro, Aveiro, Portugal

[2] Departamento de Oceanografia, Universidade Federal de Pernambuco, Recife, Pernambuco, Brazil

[3] Australian Institute of Marine Science, Townsville, Queensland, Australia

[4] Department of Mathematic & CIDMA - Center for Research and Development in Mathematics and Applications, University of Aveiro, Aveiro, Portugal

[5] Section for Applied Marine Ecology and Modelling, Department of Bioscience, Aarhus University, Roskilde, Denmark

Corresponding author
Cátia Carreira,
ccd.carreira@gmail.com

## ABSTRACT

Microbial communities distribute heterogeneously at small-scales (mm-cm) due to physical, chemical and biological processes. To understand microbial processes and functions it is necessary to appreciate microbes and matter at small scales, however, few studies have determined microbial, viral, and biogeochemical distribution over space and time at these scales. In this study, the small-scale spatial and temporal distribution of microbes (bacteria and chlorophyll *a*), viruses, dissolved inorganic nutrients and dissolved organic carbon were determined at five locations (spatial) along the Great Barrier Reef (Australia), and over 4 consecutive days (temporal) at a coastal location. Our results show that: (1) the parameters show high small-scale heterogeneity; (2) none of the parameters measured explained the bacterial abundance distributions at these scales spatially or temporally; (3) chemical (ammonium, nitrate/nitrite, phosphate, dissolved organic carbon, and total dissolved nitrogen) and biological (chl *a*, and bacterial and viral abundances) measurements did not reveal significant relationships at the small scale; and (4) statistically significant differences were found between sites/days for all parameter measured but without a clear pattern.

## INTRODUCTION

Marine bacterioplankton and phytoplankton and their associated functions are the primary controls of energy and material cycling in the global ocean. How they interact with the environment is therefore of pivotal importance for understanding ocean food web structure and biogeochemical processes (*Wiens, 1989*). Depending on the process to be studied, the scale of spatial resolution has to be adjusted accordingly. As microbes interact at the cellular level, it is essential to describe microbial community ecology at small scales (μm to cm) to capture the microbial functions and productivity in marine environments (*Azam & Malfatti, 2007*; *Stocker, 2015*). There is evidence that microbes distribute heterogeneously

at small scales in marine environments (*Azam & Long, 2001*), which has been linked to biological factors (e.g., grazing, lysis), and interactions between microbes and the environment (e.g., organic matter, aggregates) (e.g., *Seymour et al., 2006*; *Stocker et al., 2008*). Viruses are estimated to kill between 20–40% of the prokaryotic community every day, with major consequences for the microbial diversity and carbon cycling (*Suttle, 2005*). However virus-bacteria relationships are not always straightforward. Viral abundances are typically tightly coupled with bacterial abundances when large datasets are used (*Wigington et al., 2016*); however, when small datasets or small volumes are used, bacterial and viral abundances are not coupled (*Bouvy et al., 2012*; *Carreira et al., 2013*). This difference is probably a result of the time lag between infection and replication which is easier to observe at smaller scales and volumes (*Carreira et al., 2013*). Recently it has also been proposed that viral lifestyles can be inferred from viral and microbial metagenomics data (*Coutinho et al., 2017*), specifically as an indicator of predominance of the piggyback-the-winner theory. This theory asserts that during high viral and microbial abundances, viruses will switch to a lysogenic cycle, which then predominates and thus explains the negative relation between virus-microbes ratio and bacterial abundances (*Silveira & Rohwer, 2016*). It has also been demonstrated that prokaryotes can move towards a chemical cue (chemotactic behaviour), as a response to point sources of organic and inorganic matter (*Malmcrona-Friberg, Goodman & Kjelleberg, 1990*; *Hütz & Overmann, 2011*). This chemotactic behaviour has been suggested to increase the microbial degradation of dissolved organic matter (DOM) (*Fenchel, 2002*), and heterogeneous environments are suggested to have higher phytoplankton production than found under homogeneous condition (*Brentnall et al., 2003*). Such findings have implications for the way we frame marine biogeochemical cycling by microbes. Models have been used previously to understand the interaction between microbes and organic matter at small scales (e.g., *Datta et al., 2016*), while other studies have used controlled experiments (e.g., *Brumley et al., 2019*), and measured microbial distribution at small scales in a natural ecosystem (e.g., *Seymour et al., 2005*). But none of these have measured the chemical (organic and inorganic) components interacting with the microbes at small scales in a natural ecosystem.

The Great Barrier Reef (GBR) is situated on the continental shelf and slope of Australia's north-eastern coast and is the largest contiguous coral reef system in the world. The GBR has a total of ∼ 3,700 reefs which are mainly located away from shore; with the open water body separating the reef matrix from the mainland known as the GBR lagoon. The system is characterized by stable high temperatures, oligotrophic, sunlit, and alkaline waters (*Furnas et al., 2011*; *Uthicke, Furnas & Lønborg, 2014*). The microbial patchiness at the cm scale has been studied by *Seymour et al. (2005)* on one reef, demonstrating a 2- to 3.5-fold changes in the viral and bacterial concentrations over a distance of 12 cm above coral colonies. This microbial heterogeneity suggests that small-scale interactions could be important in understanding the microbial ecology and biogeochemistry of this system. But it remains to be understood how representative these single measurements are for other locations in the GBR and how this might vary over temporal scales.

In this study we determined the spatial and temporal variability in the small-scale distribution of microbes (bacteria, and chlorophyll *a*—a proxy for phytoplankton biomass),
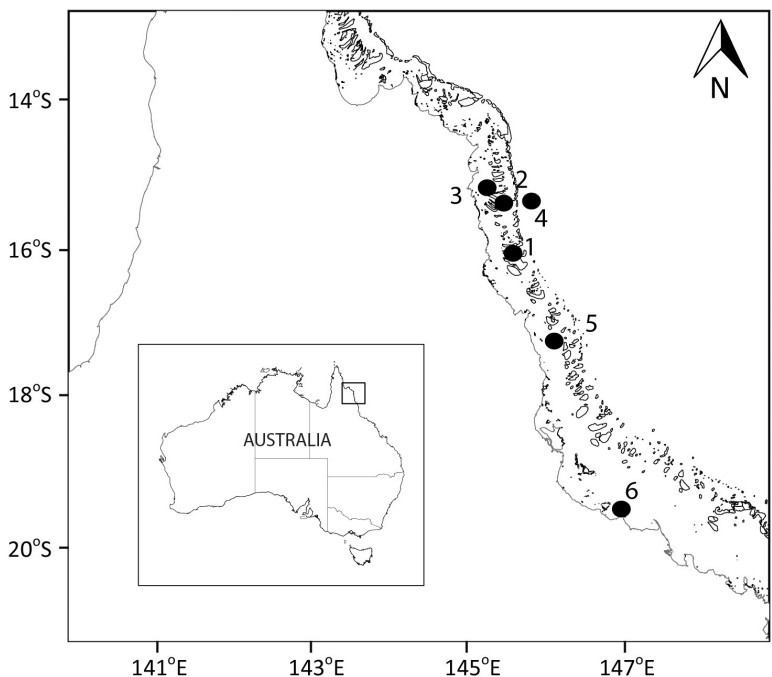

**Figure 1** **Sampling locations.** Map showing the sampled location for study sites along the Great Barrier Reef (Australia). Larger map is a representation of the square indicated in the smaller map of Australia.

and viruses, as well as biogeochemical variables (dissolved inorganic nutrients and dissolved organic carbon) at five locations along the GBR, and over 4 consecutive days at a coastal location.

Our aim was to determine microbial interactions with biological and chemical parameters using the smallest possible volume while still performing all measurements. As such, sampling at the micrometre scale ($\mu$L) where single cell interactions occur, was not possible. Instead, we sampled <25 mL, a scale that is three orders of magnitude larger than the microscale, but still three orders of magnitude lower than the typical volumes collected (litre), and it is the scale where it is possible to 'observe' microbial communities interacting ('microbial cities') (*Carreira, 2015*).

## MATERIAL AND METHODS

### Study sites and sampling

Samples were collected at six sites spanning from coast to the outer reef in the Great Barrier Reef (GBR; Fig. 1; Table 1). The Australian Institute of Marine Science (AIMS) provided a permit for this study (G37568.1). Sites 1, 2, 3, and 5 were at coral reefs, site 4 was in the Coral Sea, and site 6 was at the AIMS harbour (Bowling Green Bay; Fig. 1). Site 4 (Coral Sea) was used as a reference non-coral site. Site 6 (Bowling Green Bay) is located in the inner zone of the GBR, has no coral coverage and is dominated by a nearby saltmarsh and small river. All coral reef sites showed clumps of floating *Trichodesmium* spp. at the surface (Carreira pers. observ.) at the time of sampling. To determine the spatial variability

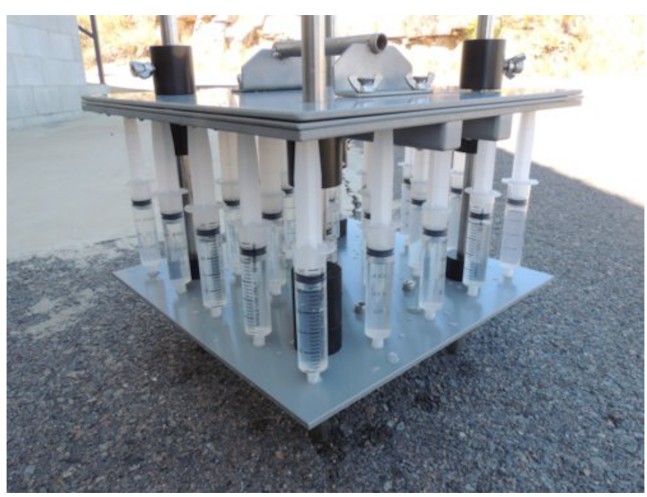

**Figure 2  Sampling device.** Two-dimensional device used for sampling consisting of 25 inlets (5 x 5) connected to a 25 mL syringe, each separated by seven cm with a total sampling area of 784 cm$^2$.

**Table 1  Sampling sites information.** Great Barrier Reef (Australia) location names, sites identification number, latitude, longitude, sampling date, time, and the in-situ salinity and temperature. Sites 1 to 5 were each sampled once for the spatial study, while site 6 was sampled over 4 days for the temporal study.

| Location | Site | Latitude | Longitude | Date | Time | Salinity | Temperature (°C) |
|---|---|---|---|---|---|---|---|
| Rudder Reef | 1 | −16.20139 | 145.76722 | 17/12/2014 | 8:00 AM | 35.5 | 30.1 |
| Irene Reef | 2 | - 15.64772 | 145.68234 | 17/12/2014 | 2:50 PM | n.d. | n.d. |
| Osterlund Reef | 3 | −15.55405 | 145.45964 | 18/12/2014 | 11:00 AM | 35.6 | 29.4 |
| Coral Sea | 4 | −15.55972 | 145.97222 | 19/12/2014 | 11:20 AM | 35.3 | 28.8 |
| Flora Reef | 5 | −17.22020 | 146.25450 | 22/12/2014 | 11:00 AM | 35.5 | 28.7 |
| Bowling Green Bay | 6 | −19.27602 | 147.05744 | 12–15/01/2015 | High Tide (2:20 PM–5:45 PM) | 32–33 | 31–32 |

in the small-scale distribution of microbes, viruses and biogeochemical measurements, surface water samples were collected once at sites 1 to 5 (Fig. 1, Table 1; 17 to 22 December 2014). The temporal variability in the small-scale distribution was determined at site 6 with samples collected during high tide every 24 h over four consecutive days (Fig. 1, Table 1; 12 to 15 January 2015). Although 24 h is not a temporal small scale, the objective was only to understand the changes in spatial small scale over time. Furthermore, although it has been shown that there can be differences in bacterial abundances on coral reefs between day and night (*Weber & Apprill, 2020*), our daily sampling over the 4 days occurred always during daylight between 2 pm and 5.45 pm (Table 1; sunset at 6.55 pm). Niskin bottle samples collected at site 6, at the same time and days as the temporal study (days 1 to 4), were used as controls for the standard sampling method.

Samples were collected with a device built for the purpose of this study consisting of 25 inlets (5 x 5) (Fig. 2). With the help of a lever, all samples were collected manually, at the same time from 0.5 m below the sea surface with the sampling taking about 5 s. As our

objective was to understand small-scale heterogeneity in the coral reef system, samples from sites 1, 2, 3, and 5 were taken above the coral reefs, but not in the proximity of a coral as done by *Seymour et al. (2005)*. Each inlet was connected to a 25 mL syringe each separated by seven cm, representing a total sampling area of 784 cm$^2$. This distance between the syringes was calculated to account the volume necessary for all measurements (25 mL) without interfering with neighbouring sampling volumes. In this calculation we assume that the rapid intake of water by the syringes is similar in shape to a turbulent jet (*Pope, 2000*). The following equations were used for the calculation:

$$V = \pi \times r^2 \times \frac{h}{3} \tag{1}$$

$$Tan\,\theta = \frac{r}{h} \tag{2}$$

$$r = \left( \frac{3 \times V \times tan\,\theta}{\pi} \right)^{\frac{1}{3}} \tag{3}$$

we combined Eqs. (1) and (2) to obtain the minimum distance ($r$) between syringes (Eq. (3), Fig. S1). Equation (1) calculates the volume of a cone ($V$), which is the water rapidly sucked up ('turbulent jet') by the syringe and Eq. (2) takes into account its shape. This allowed to calculate the distance ($r$) using a known angle of 11.8° (Eq. (3); Fig. S1) (*Pope, 2000*; *Cushman-Roisin, 2019*). This angle is always the same independent of the fluid used (*Cushman-Roisin, 2019*).

At sites 1 to 5 temperature and salinity were recorded using a conductivity-temperature-depth (CTD) profiles (Seabird SBE19Plus). At site 6 salinity samples were collected and analysed in the laboratory with a Portasal Model 8410A, while temperature was measured manually. Salinity and temperature varied between 32.0 and 35.6, and 28.7 and 32.0 °C, respectively (Table 1). From each syringe, samples were collected for dissolved inorganic nutrients (ammonium - $NH_4^+$, nitrate/nitrite - $NO_3^-/NO_2^-$, and phosphate - $HPO_4^{2-}$), dissolved organic carbon (DOC), total dissolved nitrogen (TDN), chlorophyll *a* (chl *a*), and bacterial and viral counts. A precombusted (450 °C, 4 h) GF/F filter (13 mm diameter) was used to filter five mL for inorganic nutrients analysis, and 10 mL of seawater for DOC and TDN analysis. DOC and TDN samples were fixed with 50 $\mu$L of 25% $H_2PO_4$ and kept at 4 °C, whereas inorganic nutrients were filtered and kept at −20 °C until analysed. The GF/F filters used for collecting inorganic nutrients, DOC, and TDN samples were snap-frozen in liquid nitrogen and kept at −20 °C for chl *a* extraction. All syringes, filter-holders and inorganic nutrient sample tubes were acid-washed in 10% HCl for 24 h, and then washed three times with Milli-Q water before use.

For bacterial and viral counts, unfiltered subsamples of one mL, were collected in sterile two mL Eppendorf tubes and fixed with 0.5% glutaraldehyde final concentration (25% EM-grade, Merck) for 15 min at 4 °C, after which samples were snap-frozen in liquid nitrogen and stored at −80 °C until analysis by flow cytometry (FCM).

### Samples analysis

Inorganic nutrients ($NH_4^+$, $NO_3^-/NO_2^-$ and $HPO_4^{2-}$) were determined by standard segmented flow analysis (SFA) as described in *Hansen & Koroleff (1999)*. As all the

determined $NH_4^+$ concentrations were below the detection limit of the method (<0.02 $\mu mol\,L^{-1}$) the data is not shown. The detection limit and precisions for the other parameters were: 0.02 $\mu mol\,L^{-1}$ for $NO_3^-/NO_2^-$ and 0.001 $\mu mol\,L^{-1}$ for $HPO_4^{2-}$. Please note that the $HPO_4^{2-}$ concentrations at site 2 were also below the detection limit and therefore the data is not shown. DOC and TDN concentrations were measured using a Shimadzu TOC-L carbon analyser coupled in series with a nitric oxide chemiluminescence detector according to *Lønborg et al. (2018)*. Three to five replicate injections of 150 $\mu L$ were performed per sample. Concentrations were determined by subtracting a Milli-Q blank and dividing by the slope of a daily standard curve of potassium hydrogen phthalate and glycine. Using the deep ocean reference (Sargasso Sea deep water, 2600 m) we obtained a concentration of 45.6 $\pm$ 1.8 $\mu mol\,L^{-1}$ (average $\pm$ SD) for DOC and 22.0 $\pm$ 1.5 $\mu mol\,L^{-1}$ for TDN. Please note that the TDN measurements for day 2 and 3 of the temporal study are not reported due to problems with the gas supply for the nitric oxide chemiluminescence detector during these specific sample runs. The detection limit for DOC and TDN were 8 $\mu mol\,L^{-1}$ and 0.02 $\mu mol\,L^{-1}$, and the precisions were $\pm$ 1 $\mu mol\,L^{-1}$ and $\pm$ 0.3 $\mu mol\,L^{-1}$, respectively.

Chl *a* determinations were made by extracting the GF/F filters in ethanol (96%) for 8 h. Samples were analysed spectrophotometrically according to *Strickland & Parsons (1972)*. The dectection limit and precision for the chl *a* method were 0.005 $\mu g\,L^{-1}$ and $\pm$ 0.05 $\mu g\,L^{-1}$, respectively. Flow cytometric (FCM) enumeration of bacteria and viruses was carried out using a standard bench top Becton-Dickinson FACSVerse FCM, equipped with an air-cooled argon laser (excitation 488 nm, 15 mW power) according to *Gasol et al. (1999)* and *Brussaard (2004)* for bacteria and viruses, respectively. Samples were diluted (5-60 times) in TE buffer (Tris 10 mM, EDTA 1 mM, pH 8.0), stained with SYBR Green I (Molecular Probes®; Invitrogen Inc., Life Technologies™, NY, USA) to a final concentration of $10^{-4}$ of the commercial stock solution. Bacterial samples were incubated at ambient temperature, whereas viral samples were incubated at 80 °C (*Brussaard, 2004*), both in the dark for 10 min. The trigger was set for green fluorescence and the data was analysed using Flowing Software 2.5.1. (freeware; http://flowingsoftware.btk.fi). The event rate was 300 bacteria $s^{-1}$ and between 300-800 viruses $s^{-1}$ to avoid coincidence (*Gasol et al., 1999*; *Brussaard, 2004*). We would like to note that recent research (e.g., *Forterre et al., 2013*) has suggested that viral counts might also include gene transfer agents (GTAs), membrane -derived vesicles (MVs), or even cell debris that might be confused with viruses. However currently there is no method to distinguish between all these particles, therefore, we assumed that the viral counts made by FCM are viruses.

Inorganic nutrients, DOC, TDN, and chl *a* concentration, and bacterial and viral abundances data were plotted using Surfer 9.0.

## Statistical analyses

To measure small-scale heterogeneity within each site/day it was used the coefficient of variation (CV) calculated as the (Standard deviation/Mean) $\times$100. Values closer to 0% indicate a low variability, whereas values closer to 100% indicate a high variability (*Sokal & Rohlf, 1995*).

To understand differences in concentrations/abundance between sites/days, boxplots were made using the average values for each site/day. To compare the distributions of the independent samples, the non-parametric Kruskall-Wallis tests were performed for the spatial and temporal data sets because for each variable at least one subgroup (for site or day) failed the normality condition for parametric tests (*Agresti, 2007*). Multiple comparison tests were also performed to understand which pairs of sites/days had the biggest differences. For these pairwise comparisons two tests were performed: Nemenyi tests with Chi-squared approximation and the Dunn's tests for multiple comparisons with the Bonferroni adjustment method (*Dunn, 1964*).

To identify which variables were more linearly correlated, and determine the correspondent magnitude, Pearson correlation coefficients were calculated, considering each site/day and all sites/days combined. Additionally, due to the fact in most cases the data normality was violated, we calculated Spearman correlation ($R_s$) to measure the strength of a monotonic relationship between paired data.

To understand the relation between the parameters, independent of the site and day, factor analysis was applied to the data. A factor analysis is used to describe an eventual correlation between several observed variables in regard to another group of non-observed variables, of smaller dimension, named factors (*Johnson & Wichern, 2007*). To perform the factor analysis all variables were considered, regardless of site, as there were no significant correlations between the variables, according to Bartlett's test of sphericity ($p$ value < 0.001). In order to classify the variables, cluster analyses were performed in the spatial and temporal data. For the temporal data, cluster analysis was tested, but without meaningful results. R (1.1.442) and SPSS (v25) software were used for the statistical analyses.

## RESULTS

### Small scale variability for each site and day

Using the coefficient of variability (CV) as a measure of heterogeneity, generally, in the spatial and temporal studies, there was a high small-scale heterogeneity (up to 76% for chl *a*) for chl *a*, $NO_3^-/NO_2^-$ and $HPO_4^{2-}$ and lower heterogeneity for Dissolved organic carbon (DOC), and bacterial and viral abundances. With the exception of chl *a*, the chemical variables were more variable, than the biological within each site and day (Tables 2 and 3). Next is given a description of the small-scale variability for each parameters for the sites and days measured.

In the spatial study of $NO_3^-/NO_2^-$, site 5 showed the lowest heterogeneity (10%), while site 1 had the highest (37%; Table 2, Fig. 3). In the temporal study, the highest variability was observed at day 1 (45%) and lowest at day 3 (10%) (Table 3, Fig. 4). Maximum differences observed between two nearby points in the spatial and temporal studies were of 2.6 x and 3.6 x, respectively (Figs. 3 and 4). Also site 1 showed the highest heterogeneity in $HPO_4^{2-}$ concentrations (27%), and sites 4 and 5 the lowest (20%; Table 2, Fig. 3). In the temporal study the heterogeneity of $HPO_4^{2-}$ concentrations was highest at day 4 (26%), and lowest at day 3 (10%; Table 3, Fig. 4). The maximum variability between nearby points was of 2.4 x, both spatially and temporally. DOC concentrations showed the highest

**Table 2  Spatial study data.** Total and per site average (±standard deviation, SD), minimum (Min) and maximum (Max) values for nitrate/nitrite ($NO_3^-/NO_2^-$), phosphate ($HPO_4^{2-}$), dissolved organic carbon (DOC), total dissolved nitrogen (TDN), chlorophyll $a$ (chl $a$), bacterial and viral abundances, and virus to bacteria ratio (VBR) measured at the sites 1 to 5 included in the spatial study in the Great Barrier Reef (Australia); n.d., - not determined.

| Site | Calculation | $NO_3^-/NO_2^-$ ($\mu mol\,l^{-1}$) | $HPO_4^{2-}$ ($\mu mol\,l^{-1}$) | DOC ($\mu mol\,l^{-1}$) | TDN ($\mu mol\,l^{-1}$) | chl $a$ ($\mu g\,l^{-1}$) | Bacteria (x$10^5$ ml$^{-1}$) | Viruses (x$10^5$ ml$^{-1}$) | VBR |
|---|---|---|---|---|---|---|---|---|---|
| | Average ± SD | 0.07 ± 0.02 | 0.06 ± 0.02 | 90 ± 9 | 8.4 ± 0.9 | 0.44 ± 0.21 | 7.8 ± 0.5 | 51.9 ± 3.4 | 6.7 ± 0.7 |
| 1 | Min - Max | 0.05- 0.16 | 0.04 - 0.09 | 75 - 106 | 6.9 - 10.2 | 0.11 - 0.88 | 7.3 - 9.6 | 45.6 - 60.6 | 3.5 - 7.4 |
| | CV (%) | 37 | 27 | 10 | 11 | 47 | 7 | 6 | 10 |
| | Average ± SD | 0.06 ± 0.01 | | 83 ± 11 | 6.5 ± 0.8 | 0.50 ± 0.21 | 9.6 ± 0.3 | 58.2 ± 8.8 | 6.1 ± 1.0 |
| 2 | Min - Max | 0.05 - 0.08 | n.d. | 72 - 112 | 5.3 - 8.6 | 0.14 - 0.79 | 8.8 - 10.2 | 35.6 - 72.8 | 3.5 - 7.4 |
| | CV (%) | 20 | | 13 | 12 | 43 | 4 | 15 | 16 |
| | Average ± SD | 0.08 ± 0.01 | 0.05 ± 0.01 | 85 ± 11 | 6.8 ± 1.6 | 0.51 ± 0.23 | 12.5 ± 1.9 | 47.0 ± 6.9 | 3.8 ± 0.6 |
| 3 | Min - Max | 0.05 - 0.10 | 0.04 - 0.07 | 72 - 112 | 2.1 - 10.1 | 0.06 - 1.12 | 11.2 - 20.5 | 37.7 - 67.2 | 2.8 - 4.7 |
| | CV (%) | 15 | 21 | 13 | 24 | 46 | 15 | 15 | 15 |
| | Average ± SD | 0.06 ± 0.01 | 0.07 ± 0.01 | 89 ± 8 | 6.5 ± 0.6 | 0.34 ± 0.18 | 5.5 ± 0.3 | 18.9 ± 1.1 | 3.4 ± 0.3 |
| 4 | Min - Max | 0.05 - 0.11 | 0.04 - 0.11 | 77 - 104 | 5.6 - 7.7 | 0.04 - 0.69 | 5.2 - 6.5 | 17.4 -21.4 | 2.9 - 4.1 |
| | CV (%) | 21 | 20 | 9 | 9 | 54 | 5 | 6 | 9 |
| | Average ± SD | 0.05 ± 0.01 | 0.04 ± 0.01 | 86 ± 5 | 8.1 ± 1.1 | 0.25 ± 0.17 | 10.1 ± 0.6 | 41.3 ± 2.7 | 4.1 ± 0.3 |
| 5 | Min - Max | 0.05 - 0.07 | 0.04 - 0.08 | 78 - 102 | 6.1 - 10.4 | 0.05 - 0.62 | 9.4 - 12.5 | 36.2 - 45.3 | 3.2 - 4.7 |
| | CV (%) | 10 | 20 | 6 | 13 | 68 | 6 | 7 | 9 |
| Total | Average ± SD | 0.06 ± 0.02 | 0.06 ± 0.02 | 86 ± 9 | 7.3 ± 1.3 | 0.41 ± 0.22 | 9.1 ± 2.5 | 43.4 ± 14.5 | 4.8 ± 1.5 |
| | Min - Max | 0.05 - 0.16 | 0.04 - 0.11 | 72 - 112 | 2.1 - 10.4 | 0.04 - 1.12 | 5.2 - 20.5 | 17.4 - 72.8 | 2.8 - 8.3 |

**Table 3  Temporal study data.** Total and per day average (± standard deviation, SD), and minimum (Min) and maximum (Max) values for nitrate/nitrite ($NO_3^-/NO_2^-$), phosphate ($HPO_4^{2-}$), dissolved organic carbon (DOC), total dissolved nitrogen (TDN), chlorophyll $a$ (chl $a$), bacterial and viral abundances, and virus to bacteria ratio (VBR) measured during the 4 days of the temporal study at Bowling Green Bay (site 6) in the Great Barrier Reef (Australia); n.d., - not determined.

| Day | Calculation | $NO_3^-/NO_2^-$ ($\mu mol\,l^{-1}$) | $HPO_4^{2-}$ ($\mu mol\,l^{-1}$) | DOC ($\mu mol\,l^{-1}$) | TDN ($\mu mol\,l^{-1}$) | chl $a$ ($\mu g\,l^{-1}$) | Bacteria (x$10^5$ ml$^{-1}$) | Viruses (x$10^5$ ml$^{-1}$) | VBR |
|---|---|---|---|---|---|---|---|---|---|
| | Average ± SD | 0.06 ± 0.03 | 0.08 ± 0.01 | 107 ± 4 | 8.9 ± 0.8 | 0.27 ± 0.18 | 20.3 ± 0.9 | 98.9 ± 9.2 | 4.9 ± 0.5 |
| 1 | Min - Max | 0.05 - 0.18 | 0.06 - 0.09 | 99 - 114 | 7.2 - 10.5 | 0.02 - 0.67 | 18.8 - 23.0 | 81.6 - 126.0 | 4.0 - 6.4 |
| | CV (%) | 45 | 12 | 4 | 9 | 65 | 5 | 9 | 10 |
| | Average ± SD | 0.08 ± 0.0 | 0.07 ± 0.01 | 101 ± 4 | | 0.24 ± 0.15 | 19.8 ± 2.0 | 121 .1 ± 15.7 | 6.2 ± 1.0 |
| 2 | Min - Max | 0.05 - 0.11 | 0.05 - 0.09 | 95 - 110 | n.d. | 0.01 - 0.63 | 17.8 - 25.3 | 90.2 - 155.0 | 4.4 - 8.4 |
| | CV (%) | 19 | 13 | 4 | | 63 | 10 | 13 | 16 |
| | Average ± SD | 0.18 ± 0.02 | 0.09 ± 0.01 | 118 ± 6 | | 0.16 ± 0.12 | 17.9 ± 3.3 | 111.0 ± 12.4 | 6.4 ± 1.3 |
| 3 | Min - Max | 0.15 - 0.21 | 0.07 - 0.11 | 109 - 135 | n.d. | 0.05 - 0.67 | 13.0 - 24.1 | 94.8 - 160.0 | 4.0 - 8.2 |
| | CV (%) | 10 | 10 | 5 | | 76 | 19 | 11 | 20 |
| | Average ± SD | 0.63 ± 0.09 | 0.11 ± 0.03 | 100 ± 4 | 11.3 ± 0.7 | 0.20 ± 0.10 | 14.5 ± 0.9 | 88.1 ± 10.6 | 6.1 ± 0.7 |
| 4 | Min - Max | 0.47 - 0.82 | 0.07 - 0.16 | 94 - 113 | 10.2 - 12.7 | 0.07 - 0.48 | 12.1 - 16.4 | 68.8 - 108.0 | 5.0 - 7.3 |
| | CV (%) | 15 | 26 | 4 | 6 | 52 | 6 | 12 | 12 |
| Total | Average ± SD | 0.24 ± 0.24 | 0.09 ± 0.02 | 106 ± 8 | 10.1 ± 1.4 | 0.22 ± 0.14 | 18.1 ± 3.1 | 104.8 ± 17.3 | 5.9 ± 1.1 |
| | Min - Max | 0.05 - 0.82 | 0.05 - 0.16 | 94 - 135 | 7.2 - 12.7 | 0.01 - 0.67 | 12.1 - 25.3 | 68.8 - 160.0 | 4.0 - 8.4 |

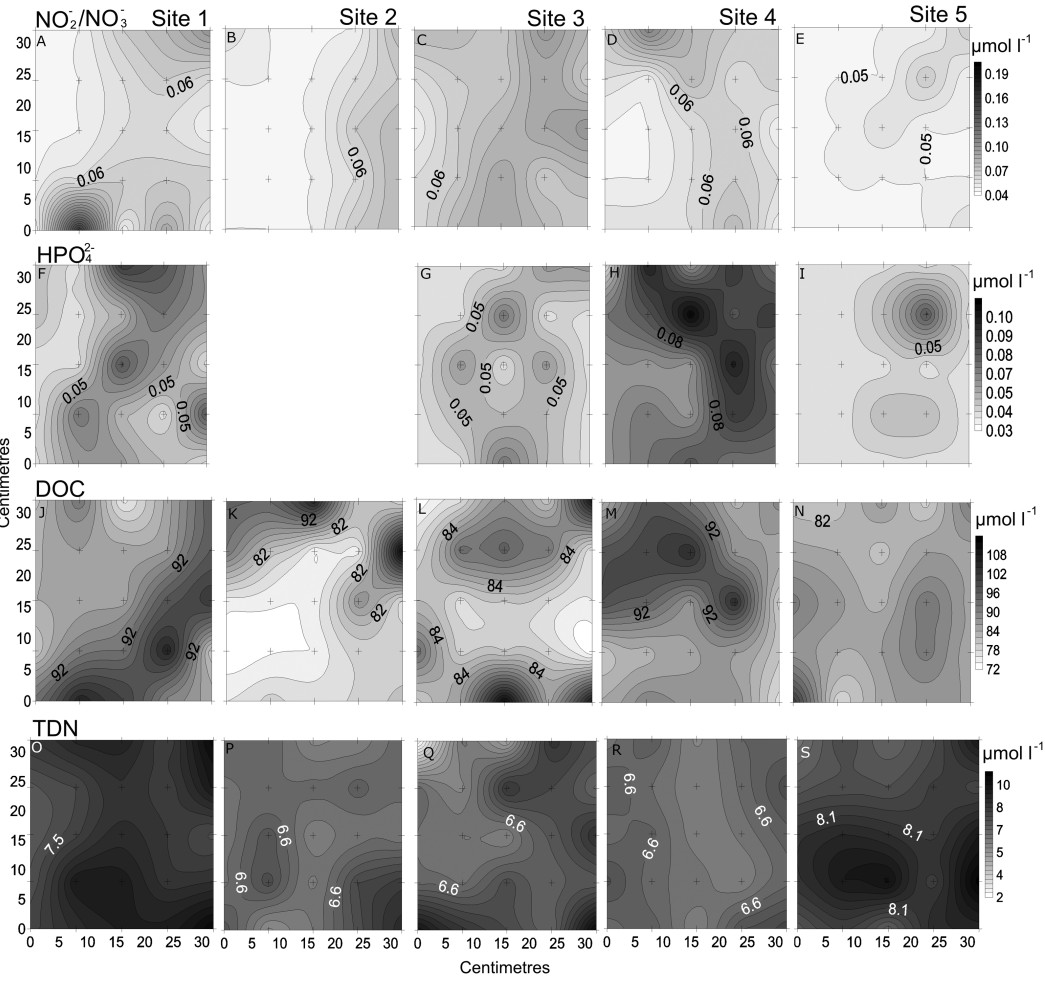

**Figure 3 Spatial distribution of chemical parameters.** Small-scale spatial distribution of nitrate/nitrite $(NO_3^-/NO_2^-$; A–E) phosphate $(HPO_4^{2-}$; F–I) dissolved organic carbon (DOC; J–N) total dissolved nitrogen (TDN; O–S; top to bottom) measured at the 5 sites (left to right) of the spatial study in the Great Barrier Reef (Australia). The grey scale represents the range of concentrations for each parameter, with white being the lowest concentration and black the highest. The axes represent the 28 cm spatial array used for sampling.

heterogeneity at sites 2 and 3 (13%; Table 2, Fig. 3), while the lowest was found at site 5 (6%). DOC showed the lowest heterogeneity of all measured parameters at all sites. The DOC concentrations in the temporal study were higher than in the spatial study, but the heterogeneity was lower. The highest heterogeneity was of 5% at day 3, and the lowest just 4% at all other days (Table 3, Fig. 4). A maximum variability of 1.5 x and 1.2 x between two nearby point was found spatially and temporally. Finally, TDN varied most at site 3 (24%) and least at site 4 (9%; Table 2, Fig. 3). A maximum variability of 3 x was found between points. In the temporal study, TDN was only measured on days 1 and 4, and the heterogeneity was low in those two days measured (9 and 6%).
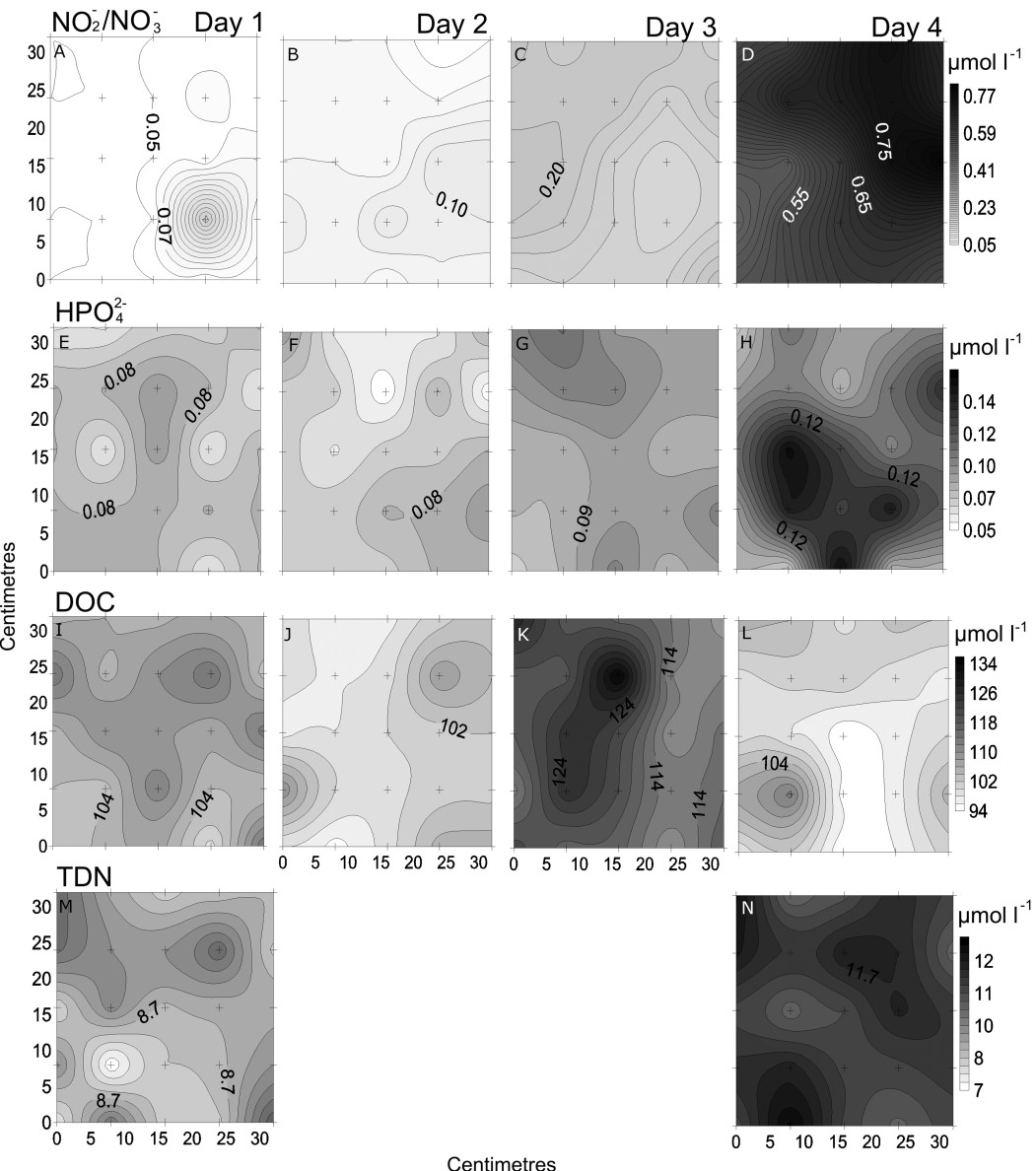

**Figure 4** **Temporal distribution of chemical parameters.** Small-scale spatial distribution of nitrate/nitrite ($NO_3^-/NO_2^-$; A–D), phosphate ($HPO_4^{2-}$; E–H), dissolved organic carbon (DOC; I–L), total dissolved nitrogen (TDN; M and N; top to bottom) measured during the 4 days (left to right) of the temporal study in the Great Barrier Reef (Australia). The grey scale represents the range of concentrations for each parameter, with white being the lowest concentration and black the highest. The axes represent the 28 cm spatial array used for sampling.

Chl *a* showed the highest variability of all parameters, with the highest heterogeneity at site 5 (68%) and the lowest at site 2 (43%; Table 2, Fig. 5). In the temporal study the heterogeneity was higher than found in the spatial study, with a maximum at day 3 (76%). Differences between nearby points were 7.9 x and 25.5 x, spatially and temporally.

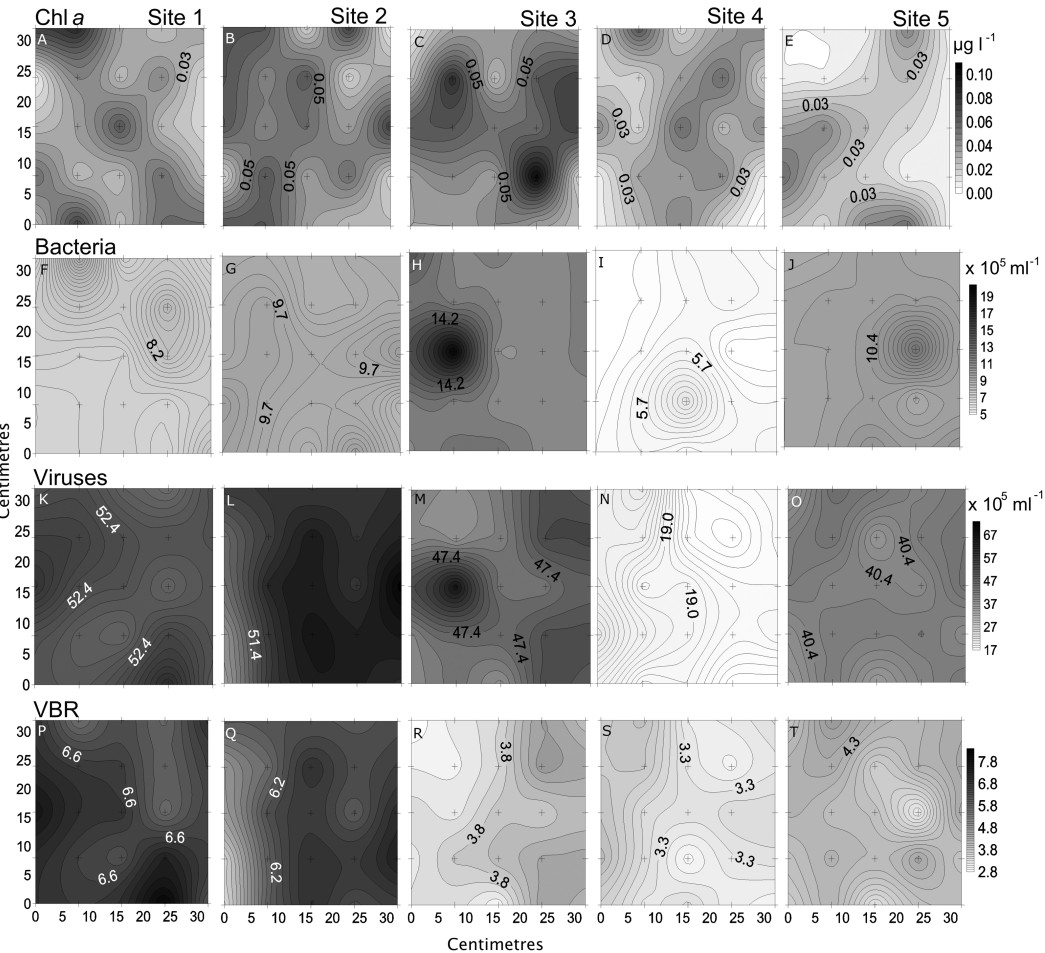

**Figure 5 Spatial distribution of biological parameters.** Small-scale spatial distribution of chlorophyll *a* (chl *a*; A–E), bacteria (F–J), viruses (K–O), and virus-to bacteria ratio (VBR; P–T; top to bottom) measured at the 5 sites (left to right) of the spatial study in the Great Barrier Reef (Australia). The grey scale represents the range of concentrations for each parameter, with white being the lowest concentration and black the highest. The axes represent the 28 cm spatial array used for sampling.

Bacterial and viral abundances showed generally similar and low heterogeneity both spatially and temporally, with viral abundances being nearly 1 order of magnitude higher than bacteria. Bacterial abundances showed similarly low heterogeneity at site 2 and day 1 (4% and 5%), and high at site 3 and day 3 (15% and 19%; Tables 2 and 3, Figs. 5 and 6). Viral abundances showed lowest heterogeneity at sites 1 and day 1 (6% and 9%), while highest heterogeneity was found at sites 2 and 3, and day 2 (15% and 13%; Tables 2 and 3, Figs. 5 and 6). Finally, the VBR showed low heterogeneity at sites 4 and 5 (9%) and day 1 (10%), and highest heterogeneity was observed at site 3 (15%) and day 3 (20%; Tables 2 and 3, Figs. 5 and 6). It should be noted that the values measured (except for nitrogen) were one order of magnitude higher than its precision, and the difference between measured values was mostly higher than the precision, thereby allowing to detect real variability between samples.

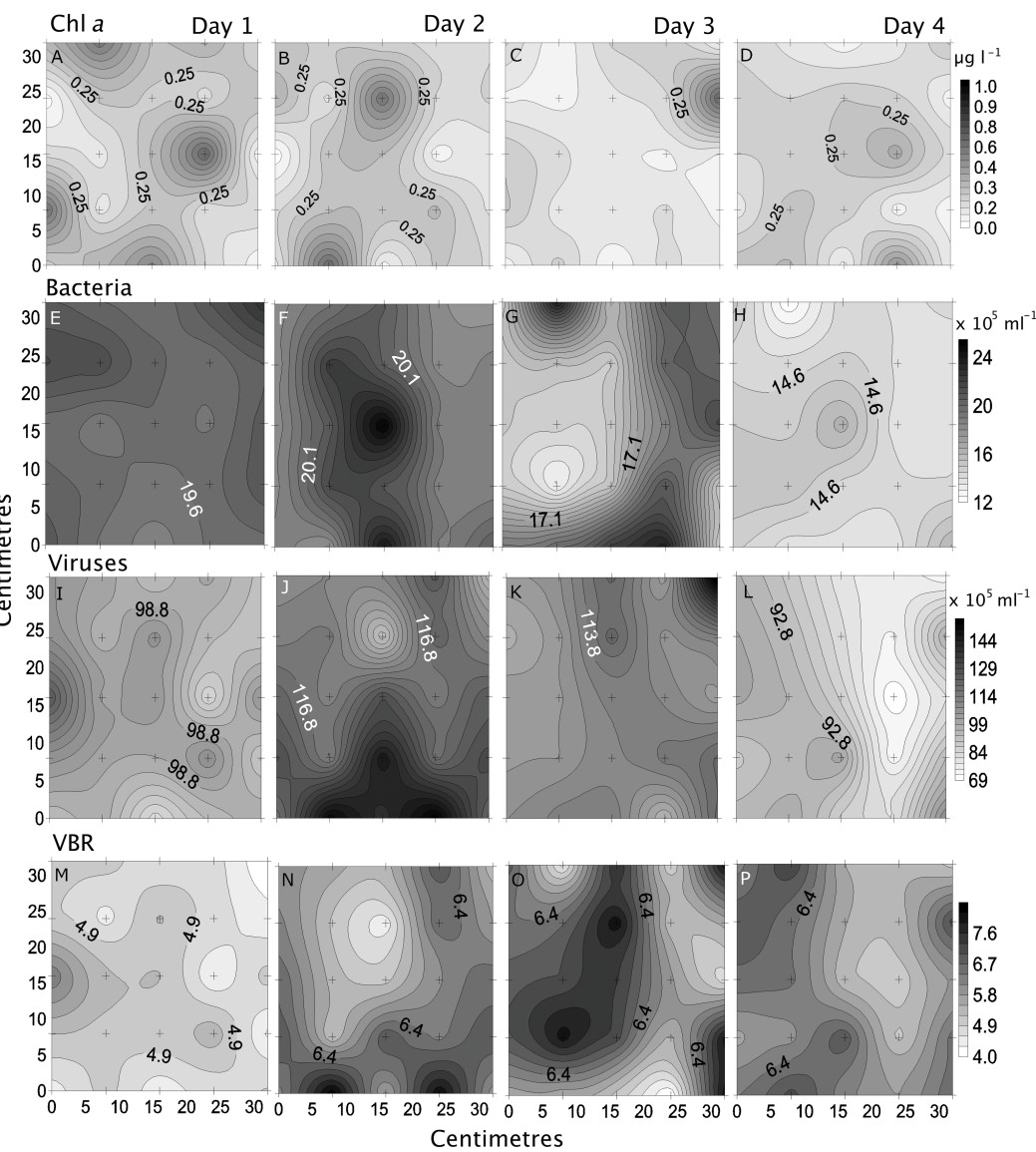

**Figure 6** **Temporal distribution of biological parameters.** Small-scale spatial distribution of chlorophyll *a* (chl *a*; A–D), bacteria (E–H), viruses (I–L), and the virus to bacteria ratio (VBR; M–P; top to bottom) measured during the 4 days (left to right) of the temporal study in the Great Barrier Reef (Australia). The grey scale represents the range of concentrations for each parameter, with white being the lowest concentration and black the highest. The axes represent the 28 cm spatial array used for sampling.

Overall, although no clear pattern emerged, site 3 (furthest north) and day 3 (high nutrient concentrations) had most parameters with highest heterogeneity, while site 5 (furthest south) and day 1 (low nutrient concentrations) had most parameters with the lowest heterogeneity.

## Comparing small-scale variability between sites and days

All sites showed comparable concentrations/abundances overall, with the exception of bacterial and viral abundances at the non-coral site 4 (Coral Sea) that were on average 1.8 x and 2.6 x lower compared to the other sites (Table 2, Fig. 5). Over the four days, nutrient concentrations increased while bacterial and viral abundances decreased, and chl *a* and VBR showed no differences (Table 3, Fig. 6, Fig. S2). The inorganic nutrients ($NO_3^-/NO_2^-$ and $HPO_4^{2-}$) showed comparable average concentrations between the outer reef and Coral Sea sites (Table 2, Fig. 3), and to site 6 (temporal study; Table 3; Fig. 4), except for day 4 when concentrations increased by 11.0 x and 1.4 x compared to day 1 (lowest concentrations, but still comparable to the sites). DOC was slightly higher at site 6 (average range over the 4 days: $100 \pm 4 \, \mu mol \, L^{-1}$ to $118 \pm 6 \, \mu mol \, L^{-1}$; Table 3, Fig. 4), compared to the coral sites (average range over sites 1, 2, 3, and 5: $83 \pm 11 \, \mu mol \, L^{-1}$ to $90 \pm 9 \, \mu mol \, L^{-1}$) and Coral Sea site ($89 \pm 8 \, \mu mol \, L^{-1}$; Table 2, Fig. 3). TDN showed slightly higher concentrations at site 6 (particularly day 4) compared to the other sites (Tables 2 and 3, Figs. 3 and 5). Chl *a* was on average lower at site 6 compared to all other sites (Tables 2 and 3, Figs. 5 and 6). Bacterial and viral abundances were on average higher at site 6 (total average of the 4 days: $18.1 \pm 3.1 \times 10^5 \, mL^{-1}$ and $104.8 \pm 17.3 \times 10^5 \, mL^{-1}$, respectively; Table 3, Fig. 6), compared to the coral sites (average range: $7.8 \pm 0.5 \times 10^5 \, mL^{-1}$ to $12.5 \pm 1.9 \times 10^5 \, mL^{-1}$; and $41.3 \pm 2.7 \times 10^5 \, mL^{-1}$ to $58.2 \pm 8.8 \times 10^5 \, mL^{-1}$ respectively; Table 2, Fig. 5). Overall site 6 (temporal study) showed either similar or slighly higher concentrations/abundances when compared to the other sites (coral sites and non-coral - Coral Sea), but these results should be taken with cautions as there was no temporal follow-up at sites 1 to 5.

Comparing the concentrations and abundances obtained with a Niskin bottles during the temporal study (Table S5) with the range of values for each parameter over the 4 days (Table 3), the values are generally within these ranges, hence also showing the increase in nutrients and decrease in bacterial and viral abundances over the 4 days.

For each variable ($NO_3^-/NO_2^-$, $HPO_4^{2-}$ DOC, TDN, Chl *a*, and bacterial and viral abundances) pairwaise comparisons between sites and days were performed. The Kolmogorov–Smirnov tests showed that the spatial and temporal data were not normally distributed for each variable. Concerning each variable in the study, the Kruskall-Wallis test revealed that at least one of the samples for each site/day is significantly different from the others. However, the pairwise comparisons using suitable non-parametric tests for multiple comparisons (Nemenyi-tests with Chi-squared approximation and Dunn's tests) showed statistically significant differences between some sites and days for each variable, but these were 'random' and without a clear link between sites/days. The *p* values are shown in the supplement material (Tables S5 and S6). Overall, statistically significant differences were found, but there were no clear patterns spatially or temporally as determined by non-parametric tests, i.e., no site and day or combinations of sites and days showed a trend or similar behaviour for all the parameters or a subsection of these (Figs. 7 and 8). DOC showed the least statistical differences between sites (Fig. 7, Table S5), while chl *a* showed the least differences between days (Fig. 8, Table S6).

Pearson and Spearman correlations were determined between all parameters within a site/day and between sites/days without any clear results (Tables S1–S4). Most correlations
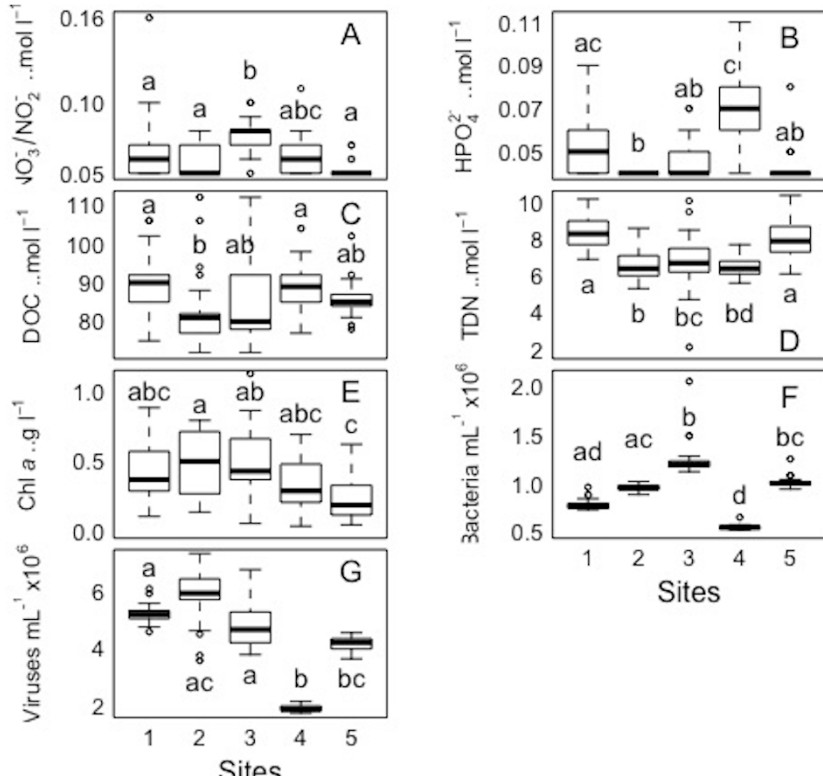

**Figure 7 Spatial distribution of biological parameters at large-scale.** Boxplots of each chemical (A-nitrate/nitrite - $NO_3^-/NO_2^-$, B-phosphate - $HPO_4^{2-}$, C-dissolved organic carbon - DOC, and D-total dissolved nitrogen - TDN), and biological parameter (E-chlorophyll $a$ - chl $a$, F-bacteria, and G-viruses) for the sampled sites (1, 2, 3, 4, and 5) in the Great Barrier Reef (Australia). Error bars represent the 10th and 90th percentiles, with 50% of the data inside the box. The solid line inside the box represents the median. Each site had a sample size of $n = 25$. Boxplots showing the same letter are not significantly different ($P < 0.05$).

did not exhibit a strong linear relationship or even a monotonic relationship between the variables in study (Tables S1–S4). However it can be highlighted that spatially, bacterial abundances correlated negatively with $HPO_4^{2-}$ and positively with viruses ($n = 25$, $R^2 = -0.54$ and 0.55, respectively), and temporally bacterial abundances correlated negatively with $NO_3^-/NO_2^-$ and $HPO_4^{2-}$ ($n = 25$, $R^2 = -0.75$ and $-0.50$). The correlation between all bacterial and all viral abundances (spatial and temporal) showed a positive correlation ($n = 325$, $R^2 = 0.75$; Fig. 9). This correlation showed an intercept not significantly different from zero, indicating a tight link between bacterial and viral abundances.

Although no relations were found between the parameters at the different sites and days, factor analysis was applied to understand the relation between the parameters. The factor analysis showed that the variables can be decomposed into two factors, the chemical ($NO_3^-/NO_2^-$ and DOC) and biological (chl $a$ and bacterial and viral abundances) groups (Fig. S3). It should be noted that $HPO_4^{2-}$ andTDN were excluded from this analysis because the correspondent anti-image matrices values were smaller than 0.5 (0.340 and

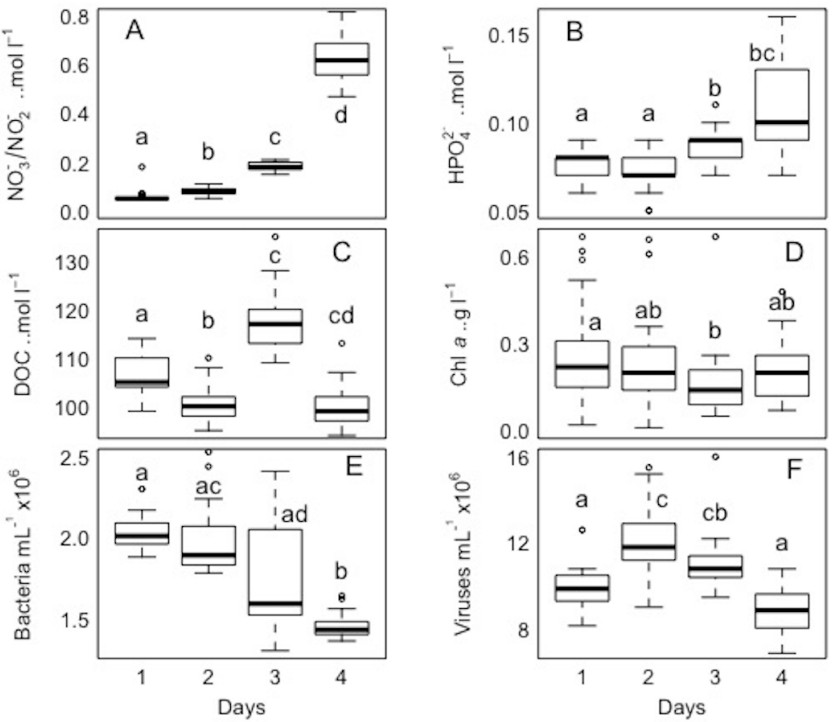

**Figure 8  Temporal distribution of biological parameters at large-scale.** Boxplots of each chemical (A-nitrate/nitrite - $NO_3^-/NO_2^-$, B-phosphate - $HPO_4^{2-}$, and C-dissolved organic carbon - DOC), and biological parameter (D-chlorophyll *a* - chl *a*, E-bacteria, and F-viruses) for the sampled days (1, 2, 3, and 4) in the Great Barrier Reef (Australia) at Bowling Green Bay (site 6). Error bars represent the 10th and 90th-percentiles, with 50% of the data inside the box. The solid line inside the box represents the median. Each day had a sample size of $n = 25$. Boxplots showing the same letter are not significantly different ($P < 0.05$).

0.372, respectively) meaning that we discarded these variables since its partial correlation values are considered too small to apply factorial analysis. Overall these results show: (1) that the chemical variables ($NO_3^-/NO_2^-$ and DOC) are more related to each others than to the biological variables (chl *a* and bacterial and viral abundances), and likewise for the biological variables, (2) the chemical variables do not explain the bacterial abundances, and (3) given the biological variables are more related to each others, there is a higher likelihood they could explain the bacterial abundances, but the results are insufficient to make a firm conclusion. Cluster analysis showed (Fig. S4A) a clear grouping between the biological (chl *a*, and bacterial and viral abundances) and the chemical ($NO_3^-/NO_2^-$, $HPO_4^{2-}$, DOC and TDN) between all sites, whereas the classification was less clear between the days (Fig. S4 B). Here bacterial and viral abundances grouped together, while DOC, chl *a*, $NO_3^-/NO_2^-$ and $HPO_4^{2-}$) clustered. Please note that the TDN data was not included in this analysis as there was no data for days 2 and 3.

The dataset generated for this study can be found at https://figshare.com/articles/Figshare_Carreira_2020Feb_xlsx/11841168.

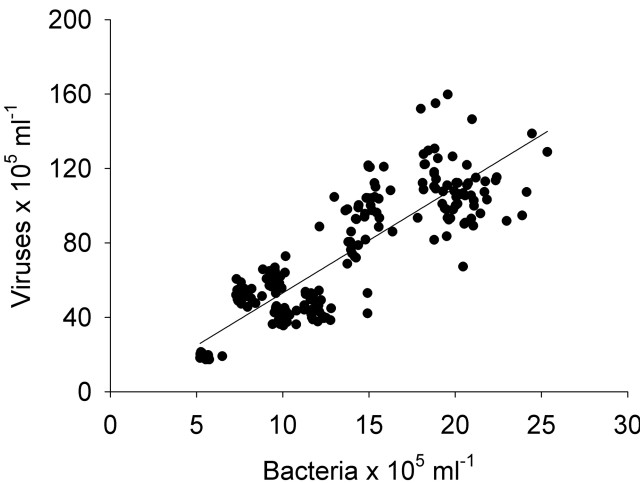

**Figure 9** **Linear regression.** Between the abundances of bacteria and viruses from all the sites (1, 2, 3, 4, and 5) and days (1, 2, 3, and 4) measured in the Great Barrier Reef (Australia) ($n = 325$; $R^2 = 0.75$; $p < 0.0001$; Vir = 5.4 ± 0.3 Bact).

## DISCUSSION

A major challenge in microbial ecology is to understand how microbial communities are influenced by changing environmental conditions. To date, however, most studies have explored these links using both larger volumes (litres) and spatial scales (km), ignoring that the water column is in fact heterogeneous at smaller scales (*Azam & Malfatti, 2007*). Previous theoretical and laboratory based studies have suggested that both microbes and their growth substrates (DOC, inorganic nutrients) have a variable distribution at small scale (*Blackburn, Fenchel & Mitchell, 1998*; *Blackburn & Fenchel, 1999*), but few studies have actually determined this combined heterogeneity in field studies (*Seymour et al., 2006*; *Weber & Apprill, 2020*). Our study shows the first in-situ heterogeneous 2-dimensional distribution of chemical (ammonium, nitrate/nitrite, phosphate, dissolved organic carbon, and total dissolved nitrogen) and biological (chl *a*, and bacterial and viral abundances) variables at the cm scale over spatial and temporal scales. Nonetheless the authors are aware that the sampled volume used in this study (ml) does not fully replicate interactions between the biogeochemistry and single microbes observed with µL samples, instead it more closely resembles interactions between microbial communities and bulk concentrations of inorganic and organic matter (*Carreira, 2015*). However this study is intended as a first approach to understand these interactions and more detailed studies at smaller scales are therefore needed. Furthermore, we have used this large dataset with 25 replicates per site/day to compare the data between sites and days.

At the resolution of our measurements none of the variables ($NO_3^-/NO_2^-$, $HPO_4^{2-}$, DOC, TDN, chl *a*, or viruses) explained the small-scale distribution of bacterial abundances at the studied sites and days (Tables S1 and S2). Nonetheless site 3 (furthest north) showed more parameters with higher heterogeneity, compared to site 5 (furthest south). At site 6 (Bowling Green Bay), day 3 showed more parameters with higher heterogeneity compared

to day 1. The increase of nutrients in Bowling Green Bay, observed at day 3, could explain the higher heterogeneity observed at this day, perhaps as a results of chemotactic behaviour by the microbes in search of food (*Malmcrona-Friberg, Goodman & Kjelleberg, 1990*; *Hütz & Overmann, 2011*).

The high variability of chl *a* (indicative of phytoplankton biomass) and nutrients found in both the spatial and temporal studies could be caused by distinct heterogeneous microenvironments created by 'Phycosphere' patches (nutrient rich areas surrounding phytoplankton cells resultant from their excretion), suggested to be hotspots for bacterial growth (*Stocker & Seymour, 2012*). However, bacterial growth was not measured in our work, and no clear link between chl *a* or nutrients and bacterial abundances was observed at the scale sampled. Additionally, the low variability observed for bacterial and viral abundances, and DOC might suggest that the sample sizes collected for analysis (1 and 10 mL) is too large to determine heterogeneity, but it could also be due to that most DOC is refractory and large proportions of cells may be dormant (*Giorgio & Scarborough, 1995*; *Lønborg et al., 2018*). Furthermore, the distribution of biological and chemical variables in the ocean are known to be impacted by processes occurring at a range of scales; for example, at the centimetre scale marine snow formation is important, while at kilometre scales fronts and eddies can shape the distribution of both chemical and biological variables (*Kiørboe, 2001*; *Jickells et al., 2008*; *Baltar & Aristegui, 2017*).

The concentrations of the chemical parameters and chl *a* were comparable to previous studies in the GBR (*Furnas et al., 2005*; *Lønborg et al., 2018*). Bacterial and viral abundances found in our study were within the estimates found for middle shelf reef surface waters in the GBR (*Alongi et al., 2015*), but about 4 - 6 and 5 - 7 x higher than those determined from a coastal coral reef in the GBR (*Seymour et al., 2005*), respectively. The Niskin bottle samples taken at site 6 (temporal study) also showed concentrations and abundances within the ranges of the small-scale sampling. However, sampling using a Niskin bottle clearly misses the high small-scale heterogeneity determined using the cm scale device from the present study. Overall, the average concentrations from the temporal study were higher than in the spatial study, which was expected as the temporal station was closer to shore.

On the whole, no pattern could be statistically detected, suggesting that the controlling factors and dynamics were different between sites and days. Some variability in the spatial study could be attributed to the differences in sampling time as our temporal study also showed differences. However the differences observed between days (Table 3; Fig. S2) are comparable to the differences observed between sites (Table 2), suggesting that differences between sites cannot solely be due to different sampling times. In the temporal study differences in the variability became more obvious over time, with nutrient concentrations increasing, while bacterial and viral abundances showed an overall decrease in abundances. A recent study by *Weber & Apprill (2020)* where microbial abundances and inorganic nutrients were followed over diel cycles in close proximity to corals (five cm) using mL samples, showed clear diel changes in the abundance of some microbial populations (specifically *Prochlorococcus* and *Synechococcus*). Furthermore, clear differences were also observed between days in the nutrients concentrations. Previous studies have also suggested that the heterogeneous distribution of microbes could be linked with
chemical (e.g., substrate), physical (e.g., turbulence) and/or biological (e.g., viral lysis) processes or a combination of these (*Stocker et al., 2008*; *Durham et al., 2013*; *Carreira et al., 2015*). A likely explanation for the spatial differences could be the variability in the quality and type of substrate, with one previous study showing spatial differences (km scale) in the concentrations of potential microbial substrate (i.e., carbohydrates and proteins) in the GBR (*Lønborg et al., 2017*). Another important factor to consider for both the spatial and temporal variability is turbulence, which increases the heterogeneity of swimming phytoplankton by 10-fold (*Durham et al., 2013*). Other studies have found spatial differences in the composition of the microbial communities (both phytoplankton and bacteria) in the GBR, which could have impacted the results in our spatial component (*Revelante, Williams & Bunt, 1982*; *Angly et al., 2016*). Grazing by microzooplankton could also have influenced the spatial and temporal variability, particularly of phytoplankton, as shown by the high mortality rates (75%) by microzooplankton of phytoplankton in tropical/subtropical regions (*Calbert & Landry, 2004*). Cell lysis might also have impacted the distributions of phytoplankton and bacteria, but we currently lack sufficient data to conclude if this is a cause for the variability found in this study. Specifically, for the temporal study, which took place at an inshore station, the daily differences in nutrient level could also have been caused by variable inputs from the nearby river and/or sediment resuspension, which both have been shown to increase nutrients concentrations in inshore parts of the GBR (*Lambrechts et al., 2010*).

The negative correlations found between nutrients ($NO_3^-/NO_2^-$ and $HPO_4^{2-}$) and bacterial abundance could be explained by a discrepancy between the timescales of nutrient uptake and bacterial growth, or bacterial growth could be limited by other factors than N and P (e.g., carbon, iron) and they therefore did not take up these nutrients (*Pinhassi et al., 2006*). No correlations were found between bacterial and viral abundances for individual sites or days ($n = 25$), but when combining all data, a relationship was observed ($n = 325$, $R^2 = 0.75$; Fig. 9). A lack of relationship between bacterial and viral abundances has previously been found in another study in a reef system with a small sample size ($n = 36$) (*Bouvy et al., 2012*), suggesting that small datasets might capture mismatched communities. This effect is then averaged out when larger datasets are pulled together (*Wigington et al., 2016*). Likewise the lack of relations between bacterial abundance and organic and inorganic nutrients could result from a discrepany between assimilation and observable changes. Thus, as most oceanographic studies collect larger samples (e.g., litres) or areas (e.g., kilometres) the interactions between microbes and viruses at small scales will not be included. Likewise, we also show that "similar" sites (reef sites) show a high degree of heterogeneity between them. Our results therefore indicate that caution is necessary when using one site or time point as a reference and it is important to consider the scale of observation to obtain an accurate understanding of microbial interactions.

Overall, the spatial study showed: (1) high small-scale variability across coral and non-coral sites, and (2) lower bacterial and viral abundances in the Coral Sea compared to coral sites. The temporal study showed: (1) persistent high small-scale heterogeneity over time, (2) 24 h is not an appropriate measure of temporal change, instead, shorter time periods should be used, and (3) Niskin bottle samples showed a similar variability, but

missed the heterogeneity observed using the device presented, and (4) variability observed across days is comparable to that across sites, hence differences between sites cannot only be attributed to different sampling times.

## CONCLUSIONS

In conclusion, this study shows that at the cm scale measured in the GBR: (1) parameters show high small-scale heterogeneity, (2) none of the parameters could explain the small-scale distribution of bacteria spatially or temporally; (3) at the scales measured no significant relation were found, and (4) statistical differences were found for the measured parameters between sites and days. As such, further studies using smaller scales than the ones used in the present study (cm), but that include biological and chemical parameters, and rates of production/degradation are needed.

## ACKNOWLEDGEMENTS

We thank the crew of the R.V. Cape Ferguson for help at sea. The help of the Cruise leader (Sven Uthicke) and participant is also acknowledged. We are thankful to Niall Jeeves for building the microsampling device and Jessica Benthuysen for the calculation of the required minimal distance between samples.

### Funding

Financial support to CESAM - The Centre for Environmental and Marine Studies (UIDP/50017/2020+UIDB/50017/2020), was given by FCT/MCTES through national funds. Cátia Carreira was supported by Fundação para a Ciência e a Tecnologia (FCT; SFRH/BPD/117746/2016). Isabel Pereira was supported by the FCT, through CIDMA - Center for Research and Development in Mathematics and Applications, within the projects UIDB/04106/2020 and UIDP/04106/2020. Financial support was provided by the Australian Institute of Marine Science. There was no additional external funding received for this study. The funders had no role in study design, data collection and analysis, decision to publish, or preparation of the manuscript.

### Grant Disclosures

The following grant information was disclosed by the authors:
CESAM: UIDP/50017/2020+UIDB/50017/2020.
FCT/MCTES.
Fundação para a Ciência e a Tecnologia (FCT): SFRH/BPD/117746/2016.
CIDMA - Center for Research and Development in Mathematics and Applications: UIDB/04106/2020, UIDP/04106/2020.
Australian Institute of Marine Science.

### Competing Interests

The authors declare there are no competing interests.

## Author Contributions

- Cátia Carreira and Christian Lønborg conceived and designed the experiments, performed the experiments, analyzed the data, prepared figures and/or tables, authored or reviewed drafts of the paper, and approved the final draft.
- Júlia Porto Silva Carvalho and Samantha Talbot performed the experiments, authored or reviewed drafts of the paper, and approved the final draft.
- Isabel Pereira analyzed the data, prepared figures and/or tables, authored or reviewed drafts of the paper, and approved the final draft.

## Field Study Permissions

The following information was supplied relating to field study approvals (i.e., approving body and any reference numbers):

The Australian Institute of Marine Science (AIMS) provided a permit for this study (G37568.1).

## Data Availability

All raw data is available at Figshare: Carreira, Cátia; Lønborg, Christian; Pereira, Isabel; Duggan, Samantha; Porto Silva Carvalho, Júlia (2020): Figshare_Carreira_2020Feb.xlsx. figshare. Dataset. https://doi.org/10.6084/m9.figshare.11841168.v1.

## Supplemental Information

Supplemental information for this article can be found online at http://dx.doi.org/10.7717/peerj.10049#supplemental-information.

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
