# Peer review of "Small-scale distribution of microbes and biogeochemistry in the Great Barrier Reef"

_PeerJ, doi:10.7717/peerj.10049_

## Round 0.1 · original submission · Major Revisions

Dear Dr Carreira,

Your manuscript has been reviewed by two external reviewers, who found that your work has merit but also suffers from some deficiencies. Upon careful consideration, I will consider evaluating a revised version that would take into account the comments and suggestions they have made.

In particular, please carefully address the following:

(1) Introduction: please reinforce the rationale and working hypotheses for the study.

(2) Sampling design: it does not really explore large-scale variations within sites, and appears not really suitable to compare across sites. The recommendation is thus to focus on the small-scale variation your experimental design mostly intends to capture.

(3) Methodology: please explore new patterns in the dataset by completing Pearson correlations by, e.g the Spearman rank correlations, to search for (non-linear) monotonic relationships. Possible new relationships identified could then be better analysed and discussed.

(4) Methodology: To what extent the -already quite large- volume sampled here provides much of a different picture than larger samples used in previous studies? The authors emphasize that there can be micron level heterogeneity, but should recognize that these could be buffered by the type of sampling used. Thus the authors should further discuss the scale(s) of heterogeneity actually explored or not in this study, in relation to instrument limitations.

(5) Make sure that all the raw data are made available.

I look forward to receiving a revised version of your manuscript along with a point by point response to the reviewers' comments and suggestions.

best regards
Xavier

Reviewer 1 ·

Basic reporting

In “Small- and large-scale distribution of microbes and biogeochemistry in the Great Barrier Reef”, Carreira and colleagues present an original study aimed at investigating the microbial ecology of reef waters at the scale of cm, which is of relevance considering that most field-based studies employ larger spatial scales. Their reporting is mostly clear and unambiguous, and the used language appropriate. I enjoyed reading the introduction and the contextual background to the study. It really shows that there is still much to be done to understand microbial variation in coastal waters, particularly in reef waters. The figures presented are relevant and of good quality. I could not find a link to the raw data. Also, the PeerJ structure including a separate conclusion section does not seem to have been followed.

Experimental design

The research questions posed by the authors are very relevant and well defined. The knowledge gap is correctly identified and justifies the need of the research. However, the experimental approach appears to be mostly preliminary and does not allow a rigorous investigation of the questions identified earlier. Whereas it is clear that the authors’ intention of studying microbial, viral and biochemical variation at the cm scale is successful, in order to study microbial variation at the larger scale a different kind of replication (including within site replication) is needed. Sites are compared at the large-scale based on a single deployment of the sampling gear per site. This totally disconsiders the actual spatial variation and heterogeneity in microbial/biochemical parameters that likely exists within site, based on habitat variation, exposure to currents and in relation to topographic features, bottom depth, etc... As such, it is not clear to me that the design actually allows the authors to compare across sites, and I would recommend they bring the focus to the small-scale variation their experimental design is intended to capture.
Methods are well described and clear and contain enough information to be replicated. However, I have some concerns with the correlative approach used to search for a linkage between biological and abiotic parameters. It is OK to use Pearson correlation coefficient and look for linear correlations (see lines 200-202 and 297-299), however an alternative approach would be to use the Spearman rank correlation coefficient to search for (non-linear) monotonic relationships. Especially because most parameters do not follow a normal distribution, it can be the case that the relationships between them are not linear. Using a non-parametric correlative approach may allow disentangling new patterns in the data.

Validity of the findings

Results are mostly inconclusive but relevant for others who may want to set up a similar experimental approach. Spatial replication was not ideal, as stated above, and the authors also recognized that their temporal replication did not allow capturing relevant patterns (for instance, diel variation). Underlying data have been provided and are sound and controlled, allowing applying statistical tests.
There is only minimal degree of speculation, and the conclusions are well stated and linked/limited to the results obtained in the study.

Additional comments

Please find some detailed comments below:

Line 25:
The authors state that “the small and large-scale spatial and temporal distribution of microbes (…) were determined”. This gives an impression that there is also a small and large-scale temporal sampling, which is not true. Please rephrase if possible.

Line 30:
Results showed “the highest average concentrations/abundances”; of what exactly?

Line 33:
Please make explicit that you are talking about abundance of bacteria and viruses in the biological measurements.

Line 34:
To which differences (which parameters) are the authors referring to in point 5?

Line 43:
Some authors could even argue that certain microbial community ecology parameters are only to be seen at the micrometer scale (see work by Roman Stocker’s lab for instance).

Line 142-145:
Is there a reason why phytoplankton densities were not determined by flowcytometry? This seems like the logical procedure since you have the instrument and the samples.

Lines 194-195:
The discussion on the application of non-parametric tests is somehow confusing. By stating “i.e., the values of each variable did significantly change over the locations” one would immediately expect to see results of a statistical test. To which values are the authors referring? Is it perhaps a matter of heterogeneity of variances here?

Lines 218-219:
Please make explicit you are talking about bacterial and viral “abundances” here and also throughout the ms.

Line 300:
Do you mean “bacteria correlated negatively with HPO4” and positively with viruses?

Lines 323-325:
This sentence reads very much like a conclusion and should therefore be removed from here.

Lines 423-432:
Both points 3) from spatial study and the temporal study do not actually read like general conclusions (they refer to a rather particular detail). I would therefore remove them from here.

·

Basic reporting

The article was written clearly and the contour plots figures provided an effective way to communicate the various datasets. I recommended some additional references in the Specific comments.

Experimental design

The experimental design and analysis methods are clearly described. Further statistical analyses should be applied to the small scale dataset.

Validity of the findings

Further discussion of caveats to sample quantity and instrument precision as this relates to the primary research questions is warranted (more details outlines below).

Additional comments

Overall comments:
The article could be improved with some minor corrections, but overall, I have no objection to publication and offer only a short list of specific comments. My biggest concern is the evaluating the differences between true environmental heterogeneity and instrument precision for measuring the concentrations of chemicals and enumerating biological entities. When examining the contour plots, I can see evidence for variation between sites and days, but not necessarily much small-scale variation. The background studies introduced in the article (e.g., Azam and Stocker studies) are referring to scales on the micron level, in which case, 25 milliliters really is not providing much of a different picture than a half-liter sample because you are still three orders of magnitude off from micron level heterogeneity. Logistically, one cannot do much with 25 ul sample, but the authors should further discuss the reality of scale in this context as it relates to their theses. Adding some hypotheses to the introduction and / or results would also help clarify the authors rationale for conducting the study.


Specific comments:
Abstract.
Line 30, highest average concentrations of what?
Line 31, Do the authors mean, none of the parameters correlated with bacterial abundances / concentration? Because you do not seem to have composition data, which would be expected in order to examine “distribution”.
Line 34, should be “small scale”, not “cm scale”
Lines 34-35, Be more specific. What differences were found between sites/days?

Introduction.
Lines 50-51. Clarify the statement: “However virus-microbes’s relationship…”
Line 51. “Viruses are typically coupled with bacterial communities…” How do you mean? Composition-wise? We know that is not true. Abundance-wise? There is much debate in the literature right now about how virus to microbe ratios indicate viral lifecycles, microbial growth rates, etc, so more references are needed here.
Reference this article regarding a microbial study on scale. L Weber, A Apprill (2020)
Diel, daily, and spatial variation of coral reef seawater microbial communities. PloS one.

Methods.
Line 100. The authors stated that samples were collected at high tide for four consecutive days. But did the time of day for sampling each site vary substantially? Reef microbes have also been shown to follow diel rhythms that were not related to tide.
Line 105. Clarify “purpose built device”.

Results and Discussion.
Lines 188 and 216. Is there a reference for this CV measurement? Is this how the authors determined that there was “high small-scale heterogeneity” as reported in the abstract?

Line 217. “high” small scale heterogeneity is subjective. There can be “more variation” or “high heterogeneity” than other measurements, but there is nothing to indicate 76% is high. Also, where are you statistics for this section (Lines 216-256)?

Lines 288-291. Can more details be provided here? “…did not point to a common pattern” seems to warrant further discussion.

Line 335. Add Apprill reference from comment above.

Please evaluate how different levels of instrument precision between the different measurement could affect your results, particularly for the small scale study.

The work represents an interesting study, nice job, just a difficult question to answer with state of the art technologies available right now!

---

## Round 0.2 · accepted · Accept

Dear authors,

I am pleased to inform you that, following the revision made based on the reviewers' comments, your manuscript is now acceptable for publication in PeerJ.

As the acceptance of a new paper on a similar subject (cited by a reviewer) has the same timing as the acceptance of your paper, I do not see it as compulsory to add this reference.

Best regards

Xavier LE ROUX

Reviewer 1 ·

Basic reporting

The reporting is clear and unambiguous and there is now a link to the dataset. The structure has also been changed to include a separate conclusion and all other guidelines have been followed. I would like to point out an interesting and very recent study dealing with large-scale variation of microbial communities across the GBR and that should be cited for updated contextualization:
Frade, P.R., Glasl, B., Matthews, S.A. et al. Spatial patterns of microbial communities across surface waters of the Great Barrier Reef. Commun Biol 3, 442 (2020). https://doi.org/10.1038/s42003-020-01166-y

Experimental design

Methods are clearly described and the addition of a non-parametric correlative approach covers my previous concerns. I am also happy with the decision of the authors to tone down the focus on their large-scale comparison.

Validity of the findings

All points are correctly addressed, all underlying data have been provided and conclusions are well stated.

Additional comments

I was pleased to read the authors' reply and that all my comments were addressed and incorporated into this new version of the manuscript. I consider the manuscript will be ready for publication after the authors include a citation to a new literature reference that has just been published (see above).
Overall, this study provides a good contribution to a field in need of new sampling/experimental approaches that can help decipher the small-scale distribution of microbes in coral reef waters.

·

Basic reporting

No comment.

Experimental design

No comment.

Validity of the findings

No comment.

Additional comments

Thank you for addressing my comments in the revised manuscript. Best wishes, Linda